# Dual-Function Semaphorin 4D Released by Platelets: Suppression of Osteoblastogenesis and Promotion of Osteoclastogenesis

**DOI:** 10.3390/ijms23062938

**Published:** 2022-03-09

**Authors:** Satoru Shindo, Irma Josefina Savitri, Takenobu Ishii, Atsushi Ikeda, Roodelyne Pierrelus, Alireza Heidari, Keisuke Okubo, Shin Nakamura, Umadevi Kandalam, Mohamad Rawas-Qalaji, Elizabeth Leon, Maria Rita Pastore, Patrick Hardigan, Toshihisa Kawai

**Affiliations:** 1Department of Oral Science and Translational Research, College of Dental Medicine, Nova Southeastern University, 3200 South University Drive, Fort Lauderdale, FL 33328, USA; sshindo@nova.edu (S.S.); rp1258@nova.edu (R.P.); aheidari@nova.edu (A.H.); de18012@s.okayama-u.ac.jp (K.O.); snakamur@nova.edu (S.N.); mrawasqa@nova.edu (M.R.-Q.); eleon@nova.edu (E.L.); mpastore@nova.edu (M.R.P.); 2Periodontic Department, Faculty of Dental Medicine, Universitas Airlangga, Surabaya 60132, Indonesia; irma-j-s@fkg.unair.ac.id; 3Department of Orthodontics, Tokyo Dental College, Tokyo 101-0061, Japan; ishiit@tdc.ac.jp; 4Department of Periodontics and Endodontics, Okayama University Hospital, 2-5-1 Shikata-cho, Kita-ku, Okayama 700-8525, Japan; aikeda.0429@okayama-u.ac.jp; 5Woody L. Hunt School of Dental Medicine, Texas Tech University Health Sciences Center El Paso, El Paso, TX 79905, USA; umadevi.kandalam@ttuhsc.edu; 6Dr. Kiran C. Patel College of Allopathic Medicine, Nova Southeastern University, 3200 South University Drive, Fort Lauderdale, FL 33328, USA; patrick@nova.edu

**Keywords:** platelets, semaphorin 4D, bone regeneration, osteoblasts, osteoclasts

## Abstract

Effects of the antiosteoblastogenesis factor Semaphorin 4D (Sema4D), expressed by thrombin-activated platelets (TPs), on osteoblastogenesis, as well as osteoclastogenesis, were investigated in vitro. Intact platelets released both Sema4D and IGF-1. However, in response to stimulation with thrombin, platelets upregulated the release of Sema4D, but not IGF-1. Anti-Sema4D-neutralizing monoclonal antibody (mAb) upregulated TP-mediated osteoblastogenesis in MC3T3-E1 osteoblast precursors. MC3T3-E1 cells exposed to TPs induced phosphorylation of Akt and ERK further upregulated by the addition of anti-sema4D-mAb, suggesting the suppressive effects of TP-expressing Sema4D on osteoblastogenesis. On the other hand, TPs promoted RANKL-mediated osteoclastogenesis in the primary culture of bone-marrow-derived mononuclear cells (BMMCs). Among the known three receptors of Sema4D, including Plexin B1, Plexin B2 and CD72, little Plexin B2 was detected, and no Plexin B1 was detected, but a high level of CD72 mRNA was detected in RANKL-stimulated BMMCs by qPCR. Both anti-Sema4D-mAb and anti-CD72-mAb suppressed RANKL-induced osteoclast formation and bone resorptive activity, suggesting that Sema4D released by TPs promotes osteoclastogenesis via ligation to a CD72 receptor. This study demonstrated that Sema4D released by TPs suppresses osteogenic activity and promotes osteoclastogenesis, suggesting the novel property of platelets in bone-remodeling processes.

## 1. Introduction

Periodontal disease is characterized by gingival inflammation with subsequent destruction of tooth-supporting soft and hard tissues [1,2]. It is well-documented that bone regeneration hardly ever occurs at the alveolar bone affected by periodontitis [3,4]. Consequently, a variety of approaches have been developed to regenerate tooth-supporting tissues, including periodontal ligament, cementum and alveolar bone [5,6,7,8]. One of these approaches includes platelet-rich plasma (PRP), a concentration of autologous platelets isolated from the plasma of patients [9]. Platelets are indeed known to be a rich source of many growth factors, such as platelet-derived growth factor, IGF-1 and TGF-β [8,10]. For this reason, platelets have been used for the regeneration of connective tissue, epithelium and tendon in orthopedic and periodontal therapies [11,12]. Some studies postulated that PRP may facilitate new bone formation at the site affected by periodontitis [13,14]. However, irrespective of the presence of robust bone regenerative factors in platelets, other studies demonstrated a lack of or limited efficacy of PRP in improving bone formation at the bone loss site in periodontitis, as well as bone defect in other oral maxillofacial pathology [15,16,17,18,19,20,21]. Based on such inconsistent clinical outcomes from the PRP-based approach to induce bone regeneration, we hypothesized that some unknown factors present in platelets might impair osteogenesis and lead to the failure of bone regeneration by PRP.

Semaphorin 4D (Sema4D), also known as CD100, is a protein belonging to the semaphorin family and was originally identified in T lymphocytes [22]. Importantly, it was recently reported that Sema4D plays a downregulatory role in bone regeneration. While Sema4D-gene knockout mice demonstrate an increased thickness and density of bone compared to wild-type mice [23,24], it also suppresses the in vitro differentiation of osteoblasts via ligation with PlexinB1 expressed on osteoblasts [24,25]. On the other hand, studies have shown that Sema4D produced by platelets plays a key role in thrombosis via platelet aggregation [26,27]. Apart from these findings, the possible effect of Sema4D produced by platelets on bone remodeling is largely unknown. Furthermore, to the best of our knowledge, no studies have ever reported on the role of Sema4D produced by platelets in periodontitis. Therefore, the present study took up this question and investigated the role of Sema4D expressed by platelets in modulating in vitro both osteoblastogenesis and osteoclastogenesis from their precursor cells.

## 2. Results

### 2.1. Activated Platelets Produce Sema4D and IGF-1

In response to stimulation with thrombin, platelets increased the level of membrane-bound Sema4D, as determined by flow cytometry (Figure 1A). According to W-blotting of supernatant of platelets, soluble Sema4D (sSema4D; 125 KD) was more detectable after stimulation with thrombin (Figure 1B; platelet number adjusted and confirmed by β-actin). ELISA for sSema4D showed that the concentration of sSema4D released in the supernatant was promoted by the stimulation with thrombin (Figure 1C). Although platelets released IGF-1, the amount produced was not affected by stimulation with thrombin (Figure 1D). These results indicate that platelets release both sSema4D and IGF-1 in the culture supernatant, whereas thrombin-mediated activation of platelets promotes the release of sSema4D, but not IGF-1.

### 2.2. Sema4D Released from TPs Downmodulates Osteoblastogenesis

Sema4D derived from osteoclasts is reported to suppress osteoblastogenesis and the mineralization process mediated by mature osteoblasts [24]. To determine the possible effect of Sema4D derived from TPs on osteoblastogenesis, anti-Sema4D neutralizing mAb [28] was employed to neutralize the activity of Sema4D released by TPs. Recombinant Sema4D and TPs suppressed both ALP activity and mineralization, as detected by alizarin red staining (Figure 2A,B). The suppression of osteoblastogenesis elicited by recombinant Sema4D, as well as TPs, was partially regained by the use of anti-Sema4D mAb (Figure 2A,B).

### 2.3. Sema4D from TPs Suppressed IGF-1-Mediated Osteoblastogenesis via the Akt/ERK Axis

The suppression of osteoblastogenesis by Sema4D is attributed to its unique ability to interrupt the IGF-1-elicited Akt/ERK signaling pathway in osteoblasts [24]. Thus, we then investigated the effect of Sema4D derived from platelets on the Akt/ERK signaling pathway using anti-Sema4D mAb. TPs activated phosphorylation of Akt and ERK in MC3T3-E1 cells. The anti-Sema4D mAb addition also induced further phosphorylation, but the anti-IGF-1 mAb addition decreased phosphorylation (Figure 3A,B). We also confirmed that the ERK inhibitor U0126 and Akt inhibitor LY294002 reduce TP-induced ALP activity (Figure 3C). The expression of mRNA for Sema4D receptors PlexinB1, PlexinB2 and CD72 was detected in the MC3T3-E1 cells (Figure 3D). The expression level of PlexinB1 and B2 mRNA were higher than that of CD72 mRNA. *Actb* was used as an internal reference control. (Figure 3D). These results suggest that Sema4D released from TPs attenuated osteoblastogenesis through suppression of an IGF-1-elicited cell signal involving the Akt and ERK pathway, which plays a pivotal role in promoting osteoblast differentiation and bone formation [29,30].

### 2.4. TPs Increase RANKL-Mediated Osteoclastogenesis and Sema4D Receptor Expression on BMMCs

The effect of TPs on RANKL-induced osteoclastogenesis was evaluated using mouse-bone-marrow-derived mononuclear cells (BMMCs) as a source of osteoclast precursors. The number of TRAP+ multinuclear cells was significantly upregulated by the exposure of RANKL-stimulated BMMCs to TPs (Figure 4A). The addition of anti-IGF-1 mAb to the coculture between RANKL-stimulated BMMCs and TPs downmodulated osteoclastogenesis (Figure 4B). Interestingly, a significantly larger amount of sSema4D (about 30-fold) was released from TPs compared to the same number of RANKL-stimulated BMMCs. Therefore, in the coculture between RANKL-stimulated BMMCs and TPs, this result indicates that larger amount of sSema4D released from TPs compared to that from RANKL-stimulated BMMCs. The expression of mRNA for Sema4D receptors PlexinB1, PlexinB2 and CD72 was determined in the RANKL-stimulated BMMCs (Figure 4D). The expression level of CD72 mRNA was significantly higher than that of PlexinB1 mRNA or PlexinB2 mRNA (Figure 4D). While the expression of Plexin B2 mRNA was detectable at a low level, we could not detect Plexin B1 mRNA. According to the immunofluorescence staining of RANKL-stimulated BMMCs (Figure 4E), the expressions of Sema4D and CD72 were detected in both multinuclear and mononuclear cells in RANKL-stimulated BMMCs. Although the expression of PlexinB2 in RANKL-stimulated BMMCs was limited in mononuclear cells, it was not detected at all in multinuclear cells. Since only limited colocalization between Sema4D and Plexin B2 and between Sema4D and CD72 was detected, it can be inferred that autocrine sSema4D may not bind to its receptors extensively.

### 2.5. Sema4D Released from TPs Increases RANKL-Mediated Osteoclastogenesis from BMMCs

Based on the results, it can be concluded that (1) TPs promote osteoclastogenesis (Figure 4A), (2) TPs produce about 30-fold more sSema4D than osteoclasts (Figure 4C) and (3) osteoclasts express receptors at a high level of CD72 and low level of Plexin B2 for Sema4D (Figure 4D,E). We then examined the possible role of Sema4D and its receptor CD72 in the TP-mediated upregulation of RANKL-induced osteoclastogenesis. As expected, the addition of TPs to RANKL-stimulated BMMCs significantly increased the number of TRAP-positive osteoclasts (Figure 5A). This was suppressed by either anti-Sema4D mAb or anti-CD 72 mAb, but not control mAb (Figure 5A). The suppressive effects of these two mAbs were also detected in the pit formation assay for RANKL-stimulated BMMCs (Figure 5B). These results indicate that sSema4D derived from TPs, via ligation with CD72, may upregulate osteoclastogenesis in the TP-exposed RANKL-stimulated BMMCs.

## 3. Discussion

The present study demonstrated that (1) intact platelets release Sema4D and IGF-1, (2) upon stimulation with thrombin, platelets increase the release of Sema4D, but not IGF-1, (3) activated platelets can suppress osteoblastogenesis through inhibition of the IGF-1-induced ERK and Akt pathway, and (4) activated platelets upregulate RANKL-mediated osteoclastogenesis via ligation of Sema4D to its receptor CD72 expressed on osteoclasts.

The finding of Sema4D released by platelets (Figure 1A–C) corroborates the results of previous studies [26,27,31,32]. We also confirmed that platelets release IGF-1 (Figure 1D), which is known as a factor that promotes osteoblastogenesis [33]. Indeed, IGF-1 is secreted by many tissues, such as liver, kidney and bone, acting as an endocrine hormone to promote bone formation [34,35]. The inhibitory effect of Sema4D on bone formation via suppression of IGF1-mediated osteoblastogenesis was reported by Negishi-Koga et al. [24]. In the latter study, Sema4D produced by osteoclasts was found to suppress the Akt-ERK signaling axis elicited by ligation of IGF-1 with its receptor expressed on osteoblasts [24]. When anti-Sema4D mAb was applied to MC3T3-E1 osteoblast precursors in the presence of thrombin-activated platelets (TPs), the present study showed that the ALP activity and mineralization, as detected by alizarin red staining, were both promoted (Figure 2B). Importantly, Akt and ERK phosphorylation induced in MC3T3-E1 cells stimulated with anti-Sema4D mAb was upregulated, suggesting that IGF-1 derived from the TP-elicited signal is downmodulated by sSema4D derived from the same platelets. Indeed, anti-IGF-1 mAb decreased Akt and ERK phosphorylation in MC3T3-E1 cells (Figure 3B). U0126 and LY294002, ERK and Akt inhibitors downmodulated the osteoblastogenesis induced in the culture of MC3T3 cells, further confirming that Akt/ERK signaling elicited by TPs is associated with osteoblastogenesis. Very interestingly, when the cell number was adjusted to the same level, TPs secreted significantly more sSema4D than RANKL-stimulated osteoclasts in vitro. We still do not know the cellular source of IGF-1 in the pathophysiological context of the bone resorption site, but in the presence of osteoclasts, our results show that the addition of activated platelets can also produce sSema4D, which downmodulates osteoblastogenesis.

A study demonstrated that activated platelets can upregulate osteoclast differentiation [36]. More specifically, RANKL-mediated osteoclastogenesis could be attributed to transforming growth factor beta (TGF-β) released from activated platelets [36]. It is true that platelets are a rich source of growth factors, including platelet-derived growth factor (PDGF), IGF-1 and TGF-β [8,10], all of which are reported to promote osteoclastogenesis [37,38,39], as well as osteoblastogenesis [40,41,42]. The present study demonstrated that IGF-1 is engaged in the induction of osteoblastogenesis (Figure 3B), as well as osteoclastogenesis (Figure 4B). Although a therapeutic strategy that attempts to block the secretion of IGF-1 may suppress osteoclastogenesis, such an approach could also attenuate IGF-1-mediated osteoblastogenesis. Furthermore, an extrinsic modulation of those growth factors may cause collateral side effects on other cells, especially those in bone-supporting connective tissue, i.e., fibroblasts and vascular cells. For this reason, the finding of the Sema4D/CD72 axis in the interaction between platelets and osteoclasts may lead to a more targeted approach to increase the bone regenerative effects of activated platelets.

TPs promoted osteoclast differentiation, as well as the resorption of minerals via possible interaction between Sema4D expressed on TPs and CD72 expressed on osteoclast precursors and mature osteoclasts. It is well-established that Sema4D binds with the high-affinity receptors Plexin B1 and B2 and low-affinity receptor CD72 [43,44,45]. The Sema4D/CD72 axis plays a crucial role in adaptive immunity, both humoral and cellular, through ligation of the lymphocyte receptor [46]. CD72 is expressed on immune cells, such as macrophages, B cells, dendritic cells and mast cells, and can mediate the immune modulatory function by ligation with Sema4D [44,47,48]. Since osteoclast precursors are macrophage-linage immune cells, it is plausible that CD72, but not Plexin B1/B2, expressed on osteoclasts functions as a pivotal receptor for Sema4D released by TPs. Owing to the presence of an immunoreceptor tyrosine-based inhibition motif (ITIM) in the CD72 receptor expressed on B cells [49], the study using CD72-KO mice indicated that CD72 may elicit an inhibitory signal that downmodulates the B cell receptor (BCR) signal [50]. Nonetheless, it remains elusive whether ligation of Sema4D with CD72 can activate ITIM. Anti-Sema4D-mAb can suppress TP-mediated osteoclastogenesis (Figure 5), but this may not correspond to the theorized negative regulatory role played by CD72. However, it is noteworthy that anti-Sema4D-mAb did promote the phosphorylation of CD72 expressed on TP-stimulated osteoclast cells (data not shown), suggesting that Sema4D/CD72 ligation may attenuate the ITIM signal. Further mechanistic studies are necessary to establish the cell signal elicited by Sema4D/CD72 in the stimulation of osteoclasts with TPs.

In conclusion, we revealed the property of Sema4D and IGF-1 expressed by activated platelets in modulating osteoclastogenesis and osteoblastogenesis. In contrast to IGF-1, which promoted both osteoclastogenesis and osteoblastogenesis, Sema4D upregulated only osteoclastogenesis, while suppressing osteoblastogenesis. In particular, the finding that anti-Sema4D mAb, as well as anti-CD72 mAb, can suppress platelet-mediated upregulation of osteoclastogenesis suggests the possible development of an antibone resorption approach that targets both Sema4D and CD72.

## 4. Materials and Methods

### 4.1. Platelet Isolation from Blood

Blood of healthy human donors was collected in a Becton Dickinson Vacutainer^®^ (containing acid–citrate–dextrose (ACD); 65 mM Na3 citrate, 70 mM citric acid and 100 mM dextrose, pH 4.4) and centrifuged at 200× *g* for 20 min to obtain platelet-rich plasma (PRP). The top two-thirds of PRP fraction were transferred into a new plastic centrifuge tube, and HEPES buffer (140 mM NaCl, 2.7 mM KCl, 3.8 mM HEPES and 5 mM EGTA, pH 7.4) was added at a 1:1 ratio (*v*/*v*), including prostaglandin E1 (PGE1, 1 µM final concentration) and apyrase (0.5 unit/mL, Sigma-Aldrich, St. Louis, MO, USA) to prevent platelet activation. In order to obtain activated platelets, Thrombin (1 U/mL, Sigma-Aldrich) was added to the platelets. Thrombin-activated platelets (TPs) were resuspended in modified Tyrode’s buffer (137 mM NaCl, 20 mM HEPES, 5.6 mM glucose, 1 g/LBSA, 1 mM MgCl_2_, 2.7 mM KCl, and 3.3 mM NaH_2_PO_4_, pH 7.4).

### 4.2. Design of Anti-Sema4D-mAb and anti-CD72 mAb

Anti-sema4D mAb and anti-CD72 mAb were generated by the methods previously reported [28]. Briefly, a specific peptide sequence of mouse Sema4D (Sema4D-peptide, KHGSCEDCVLARDPYCAWSP) and mouse CD72 (CD72-peptide, DAEQQLQACQAERAKTKENLK) were designed using a BLAST search, as well as RankPep, a Class II MHC binding site prediction tool [51]. These sequences were used as an immune antigen to generate a hybridoma producing anti-Sema4D-mAb and anti-CD72-mAb. The specificity of mAbs was confirmed by direct ELISA and Western blot using the Sema4D-peptide or the CD72-peptide as a blocking reagent.

### 4.3. Animals

C57BL/6J mice were purchased from Jackson Laboratory (Bar Harbor, ME, USA). Experimental procedures using mice were approved by the Forsyth Institutional Animal Care and Use Committee (IACUC). Some bone marrow cells were also isolated from mice under the protocol approved by IACUC at Nova Southeastern University. This study was performed in accordance with ARRIVE guidelines for preclinical animal studies.

### 4.4. Mouse Bone Marrow Cell Culture and Osteoclast Differentiation

Bone-marrow-derived mononuclear cells (BMMCs) isolated from femur and tibia of C57BL/6J mice were seeded in a 96-well plate (1 × 10^5^ cells/well) in α-modified minimal essential medium (α-MEM) (Thermo Fisher Scientific, Waltham, MA USA) supplemented with 10% fetal bovine serum (FBS) (R&D Systems, Minneapolis, MN, USA). After preincubation of bone marrow cells with macrophage colony-stimulating factor (M-CSF) (R&D Systems) for 3 days, BMMCs were further cultured with or without anti-Sema4D monoclonal antibody (mAb) (50 µg/mL), anti-CD72 mAb (50 µg/mL) or control mAb (50 µg/mL) in the presence of M-CSF (50 ng/mL) and RANKL (100 ng/mL) (BioLegend, San Diego, CA, USA) and presence or absence of TPs. Osteoclasts differentiated in the culture were identified by staining with tartrate-resistant acid phosphatase (TRAP) using the Acid Phosphatase Leukocyte TRAP staining kit (Sigma-Aldrich) at Day-7. TRAP-positive cells, containing 3 nuclei or more, were counted microscopically as mature osteoclasts.

### 4.5. Pit Formation Assay

BMMCs were cultured with M-CSF for 3 days, followed by stimulation with M-CSF and RANKL in the presence or absence of anti-Sema4D mAb, anti-CD72 mAb or control mAb in a 96-well Osteo Assay Surface plate (Corning, Corning, NY, USA). After 7 days, the plates were washed with sodium hypochlorite and air-dried. Wells were imaged with a 4× objective using the Evos cell imaging system (Thermo Fisher Scientific). Image analysis was carried out with ImageJ software (version 1.53f 25 software, Bethesda, Montgomery, MD, USA).

### 4.6. Osteoblastogenesis Assay

MC3T3-E1 cells (ATCC, Manassas, VA, USA) were incubated in a 24-well plate (1 × 10^5^ cells/well) and cultured in Basal Medium supplemented with 50 μg/mL ascorbic acid and 5 mM β-glycerophosphate in the presence or absence of anti-Sema4D mAb (50 µg/mL) or control IgG (50 µg/mL) and in the presence or absence of TPs or recombinant human Sema4D (Abcam, Waltham, MA, USA).

After 7 days, Alkaline Phosphatase (ALP) staining was performed with Alkaline Phosphatase (ALP) Staining kit (Sigma-Aldrich) following the manufacturer’s instruction. After 21 days, Alizarin Red S (Sigma-Aldrich) staining was performed. In another experiment, MC3T3-E1 cells were cultured with U0126 (ERK inhibitor; 10 μM, Thermo Fisher Scientific) or LY294002 (Akt inhibitor; 10 μM, Thermo Fisher Scientific). After 7 days of culture, the cells were stained with ALP Staining kit.

### 4.7. Real-Time RT-PCR

After stimulation of BMMCs with M-CSF and RANKL/M-CSF for 24 h or MC3T3-E1 cells, total RNA was extracted from the cells using Trizol (Thermo Fisher Scientific) and reverse-transcribed using the Verso cDNA Synthesis Kit (Thermo Fisher Scientific) in the presence of random primers and oligo (dT)**.** The cDNA was mixed with Real-time PCR Master Mix (Roche, Basel, Switzerland), SYBR Green (Roche) and matched primers and then subjected to real-time RT-PCR on a Light Cycler System (Roche). All values were normalized with respect to β-actin mRNA level in each sample, and the results are expressed relative to corresponding value in M-CSF alone. The following primers were used in this study:

PlexinB1 Forward:5′-CCCTCGGTCTCCGGGTAAG-3′ 

     Reverse:5′-CATGACCTGAGCAGGAGTCAC-3′

PlexinB2 Forward:5′TGGTTCCTGCTGTAGCCATC-3′ 

     Reverse:5′-GATGTCTCCGTGCTTCCTGA-3′

CD72 Forward: 5′-CTGCACATCTCTGTCCTCCA-3′

     Reverse 5′-TCAGAGTCCTGCCTCCACTT-3′; 

β-actin Forward:5′-CTAAGGCCAACCGTGAAAAG-3′

     Reverse: 5′-ACCAGAGGACTACAGGGACA-3′. 

Simultaneously amplified β-actin gene was used as an internal control. 

### 4.8. Flow Cytometry

Platelets were stimulated with/without thrombin (0.5 or 1 unit/mL). Platelets were stained with anti-CD61-conjugated Alexa 647 (BioLegend) and anti-Sema4D-conjugated PE (BioLegend) and analyzed by FACSAria II (BD Biosciences, Franklin Lakes, NJ, USA) and FlowJo v8 software (BD Biosciences).

### 4.9. Enzyme-Linked Immunosorbent Assay (ELISA)

Sema4D and IGF-1 were measured with a modified ELISA, using the following protocol [28]. Briefly, culture supernatants were biotinylated with EZ-Link-Sulfo-NHS-Biotin (Thermo Fisher Scientific) and then reacted with immobilized anti-Sema4D-antibody (R&D Systems) or anti-GF-1-antibody (Sigma-Aldrich). After washing the plate, horseradish-peroxidase-conjugated streptavidin was reacted with the sample-derived biotin-labeled Sema4D and IGF-1. Biotin-labeled recombinant human Sema4D and IGF-1 were used as a standard control for semi-quantification of Sema4D and IGF-1.

### 4.10. Western Blotting

MC3T3-E1 cells were stimulated with anti-Sema4D mAb (50 μg/mL), anti-IGF-1 mAb (1 μg/mL) or control IgG in the presence of platelets. The cells were lysed in lysis buffer (1 M tris-HCl pH: 7.8, 2.5 M NaCl, 0.5 M EDTA pH 8.0, 10% NP-40, 2.5% sodium deoxycholate (SD) and 20% SDS) supplemented with protease inhibitor cocktail (EMD Millipore, Burlington, MA, USA) for 1 h on ice after indicated time stimulation. The lysates were centrifuged at 15,000 rpm for 15 min and heated for 10 min at 70 °C before being loaded on a gel and separated by sodium dodecyl sulfate polyacrylamide gel electrophoresis (SDS-PAGE). After transferal to a hydrophilic polyvinylidene fluoride (PVDF) membrane, the blot was probed overnight at 4 °C with mouse anti-Sema4D, rabbit anti-β-actin monoclonal antibody (Cell Signaling Technology, Danvers, MA, USA), rabbit anti-total ERK monoclonal antibody (Cell Signaling Technology), rabbit anti-phosphorylated ERK monoclonal antibody (Cell Signaling Technology), mouse anti-total Akt monoclonal antibody (Cell Signaling Technology) or mouse anti-phosphorylated Akt monoclonal antibody (Cell Signaling Technology). After 1 h of incubation at room temperature with HRP-labeled anti-mouse immunoglobulin antibody (Cell Signaling Technology) and HRP-labeled anti-rabbit immunoglobulin antibody (Cell Signaling Technology), signals were visualized using ECL Prime detection reagent (GE Healthcare, Chicago, IL, USA). Band density of blots was measured using ImageJ software (version 1.50).

### 4.11. Histological Analysis

For immunocytochemistry, BMMCs (1 × 10^6^) were grown on sterilized round cover glasses (Matsunami Glass, Osaka, Japan) with M-CSF and RANKL in a 24-well plate. BMMCs were fixed with 3.7% formaldehyde for 10 min, blocked with 1% BSA in PBS for 1 h at room temperature and then incubated with anti-Sema4D mouse monoclonal antibody conjugated with FITC (BD) anti-PlexinB2 mouse monoclonal antibody conjugated with PE (BD), anti-CD72 mouse monoclonal antibody conjugated with PE (BD Biosciences), Alexa 647 Phalloidin (Thermo Fisher Scientific) and DAPI (BD Biosciences) overnight at 4 °C. The next day, cells were embedded in Fluoromount-g (Thermo Fisher Scientific). Immunofluorescence signals were observed using the Zeiss LSM780 Confocal Microscope (Carl Zeiss, Dublin, CA, USA).

### 4.12. Statistical Analysis

Student’s *t*-test was used for comparison of two different outcomes of experiments performed. Nonparametric data were evaluated using the Mann-Whitney U test or one-way ANOVA for two-group comparisons. *p* value < 0.05 was considered statistically significant.

## Figures and Tables

**Figure 1 ijms-23-02938-f001:**
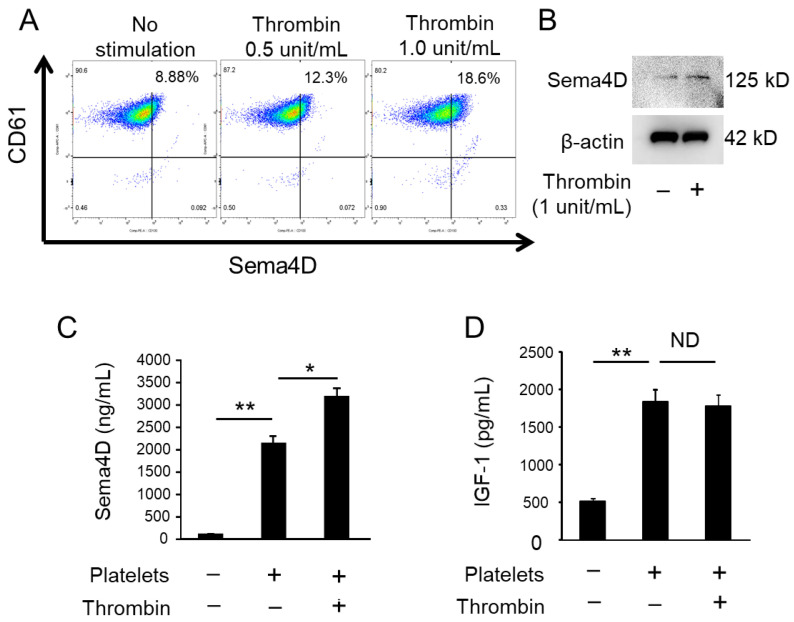
**Sema4D production from TPs.** Flow cytometry was performed to detect Sema4D on CD61+ platelets stimulated with and without thrombin (0.5, 1 unit/mL) (**A**). Sema4D and β-actin were detected in platelets cultured with/without thrombin (1 unit/mL) by Western blotting (**B**). Sema4D and IGF-1 in supernatant of platelets cultured with and without thrombin (1 unit/mL) were detected by ELISA (**C**,**D**). Data represent the mean ± SD of three independent experiments. Values are mean ± S.D. * *p* < 0.05, ** *p* < 0.01. ND: No statistically significant difference. +: Applied. −: Not applied.

**Figure 2 ijms-23-02938-f002:**
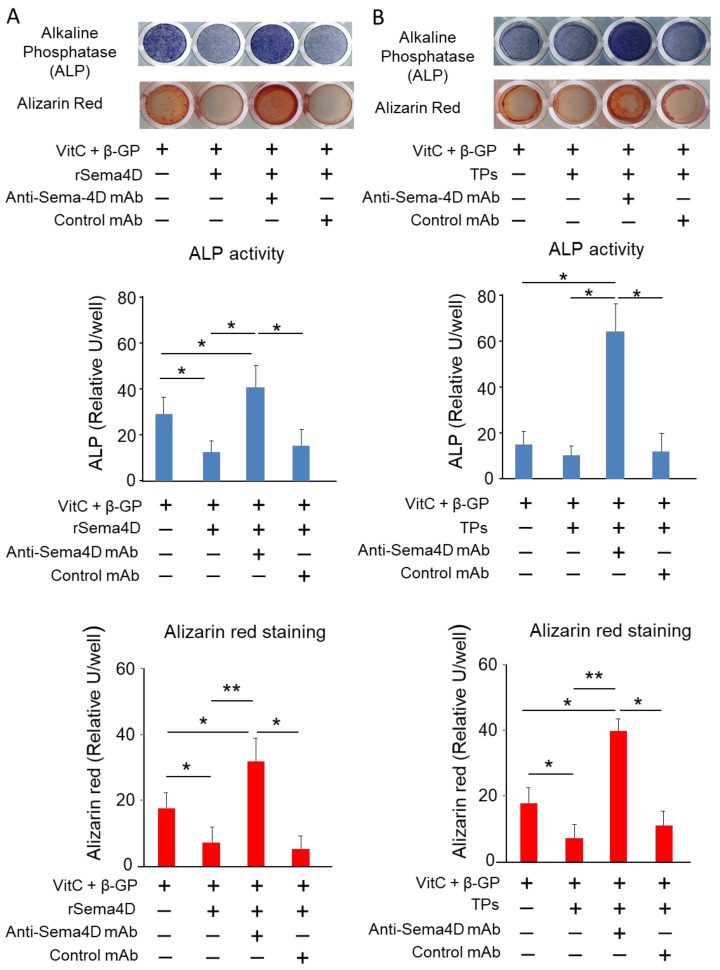
**The effect of Sema4D-neutralized TPs on osteoblastogenesis.** MC3T3-E1 cells were cultured with and without recombinant human Sema4D (10 ng/mL). After 7 days, ALP staining was performed, and after 21 says, alizarin red staining was performed (**A**). MC3T3-E1 cells were cultured with and without platelets (1 × 10^4^ cells) and anti-Sema4D antibody (50 μg/mL) or control IgG (50 μg/mL). After 7 days, ALP staining was performed, and after 21 days, alizarin red staining was performed (**B**). Data represent the mean ± SD of three independent experiments. Values are mean ± S.D. * *p* < 0.05, ** *p* < 0.01. +: Applied. −: Not applied.

**Figure 3 ijms-23-02938-f003:**
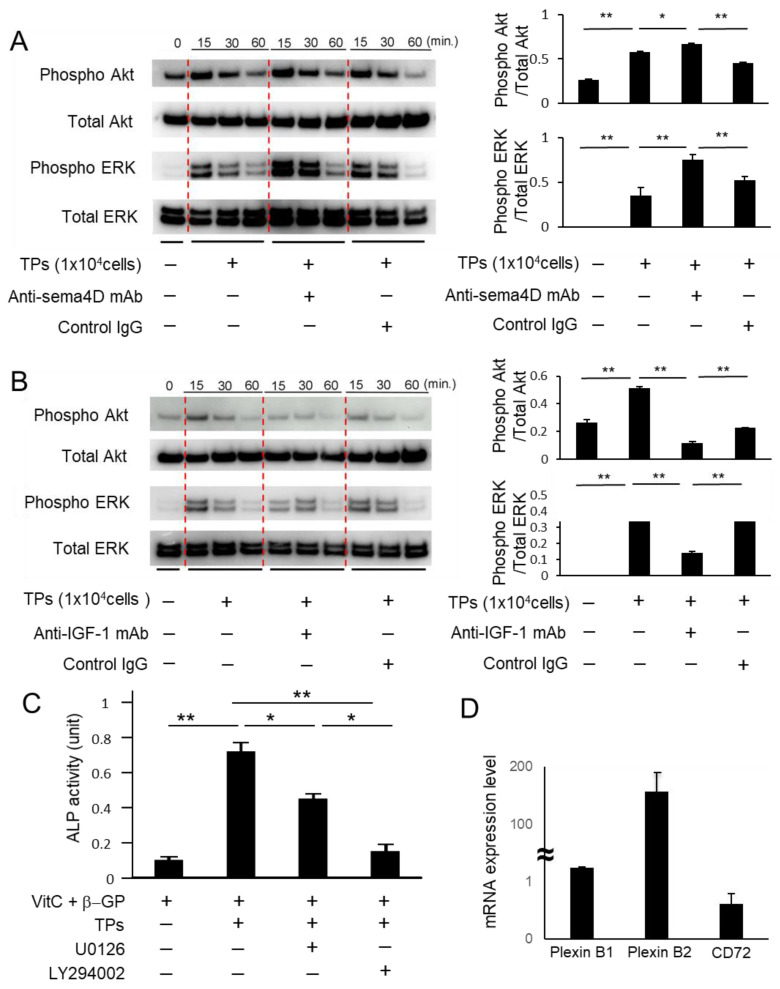
**The effect of Sema4D and IGF-1 derived from TPs on ERK and Akt phosphorylation in MC3T3-E1 cells.** Western blotting was performed to detect total and phosphorylated ERK and Akt in MC3T3-E1 cells cultured with/without ant-Sema4D mAb (50 µg/mL) or control mAb (50 µg/mL) in the presence of TPs (**A**) and total and phosphorylated ERK and Akt in MC3T3-E1 cells cultured with/without ant-IGF-1 mAb (1 µg/mL) in the presence of TPs (**B**). ALP activity was measured in TP-mediated MC3T3-E1 cells with and without ERK or Akt inhibitor (**C**). The mRNA expression of PlexinB1, PlexinB2 and CD72 in MC3T3-E1 cells was evaluated by qPCR (**D**). Data represent the mean ± SD of three independent experiments. Values are mean ± S.D. * *p* < 0.05, ** *p* < 0.01. +: Applied. −: Not applied.

**Figure 4 ijms-23-02938-f004:**
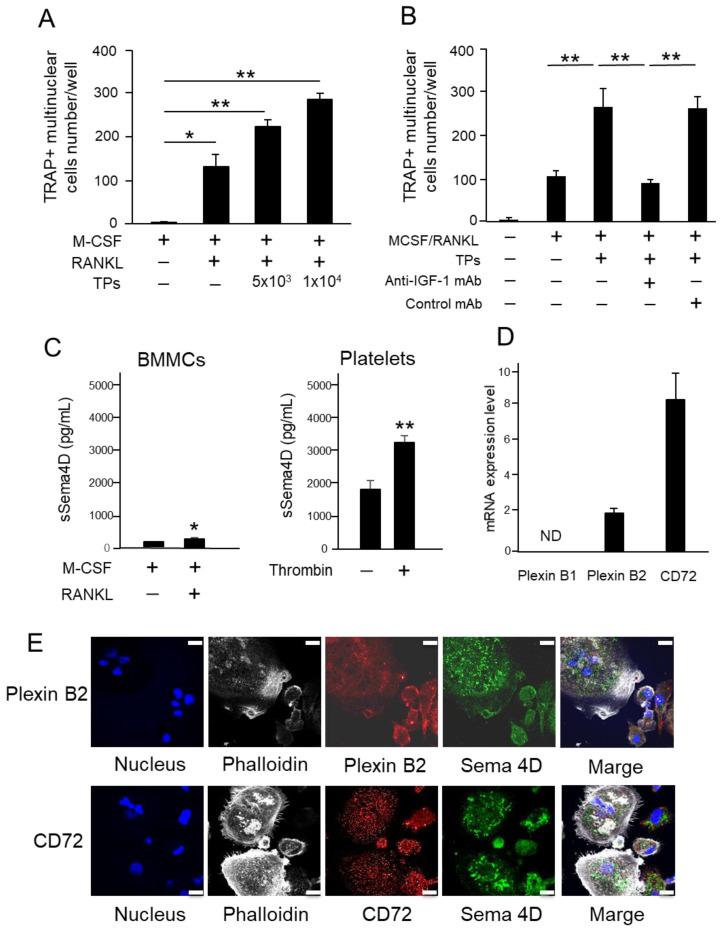
**The effect of TPs on osteoclastogenesis and the expression of receptor of Sema4D in BMMCs.** BMMCs were cultured in osteoclastogenic medium (Basal medium supplemented with M-CSF/RANKL) with or without TPs (5 × 10^3^ and 1 × 10^4^ cells). The number of TRAP-positive multinuclear cells (over three nuclei) were counted (**A**). BMMCs were stimulated with TPs (1 × 10^4^ cells) in the presence or absence of anti-IGF-1 mAb or control mAb. The number of TRAP-positive multinuclear cells (over three nuclei) were counted (**B**). ELISA was carried out to detect sSema4D in supernatant of M-CSF/RANKL-stimulated BMMCs or thrombin-stimulated platelets (**C**). The mRNA expression of PlexinB1, PlexinB2 and CD72 in BMMCs stimulated with MCSF/RANKL for 24 h was monitored by qPCR, whose level was then expressed in comparison to the housekeeping gene, *Actb* (**D**). The protein expression of PlexinB2 and CD72 was observed through immunofluorescence staining. The images in the first row are indicated as “PlexinB2” while those in the bottom row are indicated as “CD72”. Blue (DAPI): Nuclei, White (Alexa 647): Phalloidin, Red (PE): Plexin B2 or CD72, and Green (FITC): Sema4D. Bar = 10 µm. (**E**). Data represent the mean ± SD of three independent experiments. Values are mean ± S.D. * *p* < 0.05, ** *p* < 0.01. ND: No statistically significant difference. +: Applied. −: Not applied.

**Figure 5 ijms-23-02938-f005:**
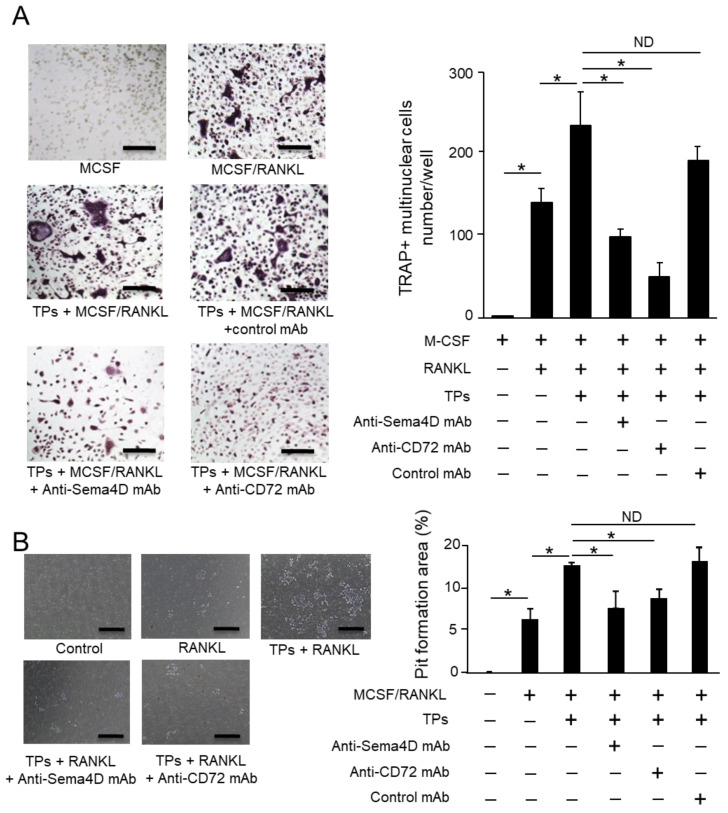
**The effect of anti-Sema4D mAb and anti-CD72mAb on RANKL-dependent osteoclastogenesis.** BMMCs were cultured in osteoclastogenic medium and TPs (1 × 10^4^ cells) with or without anti-Sema4D mAb, anti-Cd72 mAb or control mAb. The number of TRAP-positive multinuclear cells (over three nuclei) was counted (**A**), and pit formation area was evaluated (**B**). Data represent the mean ± SD of three independent experiments. Values are mean ± S.D. * *p* < 0.01. ND: No statistically significant difference. Bar = 50 µm. +: Applied. −: Not applied.

## Data Availability

The data presented in this study are available on request from the corresponding author.

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
