# Peer review of "Dual-Function Semaphorin 4D Released by Platelets: Suppression of Osteoblastogenesis and Promotion of Osteoclastogenesis"

_ijms, 2022, doi:10.3390/ijms23062938_

Round 1

Reviewer 1 Report

In this manuscript from Satoru Shindo and colleagues, the authors investigate the effects of semophorin 4D on osteoblasts and osteoclasts in culture. This represents an important and interesting topic for investigation. The research presented in this paper appears to have been performed and presented very well.

Interest in this topic stems from generally unsuccessful attempts to use platelet-rich plasma to promote periodontal bone regeneration. The authors hypothesize that semaphorin 4D produced by platelets may regulate osteoblasts and osteoclasts as one of the factors impacting effects of PRP on bone healing. Key findings are that thrombin increases expression of Sema4D by platelets, platelet-derived Sema4D inhibits differentiation of MC3T3 osteoblasts and enhances differentiation of osteoclasts from bone marrow monocytes. These data suggest that that platelets might be important modulators of bone remodeling and repair via semaohorin 4D.

The experiments performed in this study are well-designed, data presented are clear and the conclusions are reasonably derived from their results. The writing of the paper is lucid and straightforward. No significant weaknesses were noted by this reviewer.

Reviewer 2 Report

The manuscript « Dual-function Semaphorin 4D (Sema4D) released by platelets: promotion of osteoclastogenesis and suppression of osteoblastogenesis » describes effects of Sema4D released by thrombin-activated platelets (TPs) on osteoblastogenesis and osteoclastogenesis, respectively using mouse MC3T3-E1 cells or mouse bone marrow-derived mononuclear cells (BMMCs).

Authors demonstrated that Sema4D is released by platelets following stimulation with thrombin. They used a neutralizing antibody direct against Sema4D to block partially the suppression of osteoblast differentiation of MC3T3-E1 cells by recombinant or TPs-secreted Sema4D. Activation of ERK signaling pathway by IGF-1 was attenuated by Sema4D resulting in inhibition of osteoblast differentiation.

Concerning this part of results (2.1 to 2.3), the expression of potential Sema4D receptors (CD72, PlexinB1 and B2) was not described in MC3T3-E1 cells. Authors may add expression results in MC3T3-E1 cells and comments on what is known concerning Sema4D receptors in other osteoblasts.

Authors have tested the effect of Sema4D released by TPs on RANKL-mediated osteoclastogenesis using BMMCs and investigated the expression of Sema4D receptors.

In the section title 2.4, authors indicated that “TPs increase RANKL-mediated osteoclastogenesis via ligation of Sema4D with CD72, but not PlexinB1or B2” (lines 131-132). However they were less affirmative in interpreting the results, “Since only limited colocalization between 151 Sema4D and Plexin B2 and between Sema4D and CD72 was detected, it can be inferred 152 that autocrine sSema4D may not be binding to its receptors extensively.” (lines 151-153). Authors may consider changing this section title by “TPs increase RANKL-mediated osteoclastogenesis and Sema4D receptor expression on BMMCs”. Moreover anti-Sema4D was not used in this experiment, while anti-IGF-1 was used. For similar reasons, authors may consider changing the section title 2.5 “Sema4D released from TPs increases RANKL-mediated osteoclastogenesis from BMMCs via possible interaction with its low-affinity receptor CD72” (line 179) by “Sema4D released from TPs increases RANKL-mediated osteoclastogenesis from BMMCs”. The possible interaction of Sema4D with its low-affinity receptor CD72 is an important part of the discussion.

Authors may consider changing the title of their manuscript following the order that has been chosen to present the results: « Dual-function Semaphorin 4D (Sema4D) released by platelets: suppression of osteoblastogenesis and promotion of osteoclastogenesis ».

Minor revisions:

  • Number of experiments should be indicated for each figure presenting statistical analyses. Means +/- SD of how many experiments?
  • Increase the size of police in figure 3.
  • Change police style for ITIM (line 266) and BCR (line 268).
  • Use directly the abbreviation TPs (line 139).
  • Please change “as osteoclast precursors” line 134 by “as a source of osteoclast precursors”. Indeed stromal cells may still be present during this osteoclastogenesis test. Monocytes have been neither isolated by a specific antibody nor characterized by flow cytometry following cell culture.
  • Change legend for figure 4D: “mRNA expression level” relative to what (house keeping gene; culture condition) ?
  • Change legend for figure 4E: upper-line was indicated as “Plexin B2” while bottom-line was indicated as “CD72”. Should it be replaced by “mononuclear cells” and “multinuclear cells”, respectively?
  • Remove sentences lines 191-194 “This section may be divided…that can be drawn”.

The present manuscript is interesting but needs some improvements.
